# Revisit Fuzzy Neural Network: Demystifying Batch Normalization and ReLU with Generalized Hamming Network

**Lixin Fan**
`lixin.fan@nokia.com`
Nokia Technologies
Tampere, Finland

## Abstract

We revisit fuzzy neural network with a cornerstone notion of *generalized hamming distance*, which provides a novel and theoretically justified framework to re-interpret many useful neural network techniques in terms of fuzzy logic. In particular, we conjecture and empirically illustrate that, the celebrated batch normalization (BN) technique actually adapts the "normalized" bias such that it approximates the rightful bias induced by the generalized hamming distance. Once the due bias is enforced analytically, neither the optimization of bias terms nor the sophisticated batch normalization is needed. Also in the light of generalized hamming distance, the popular rectified linear units (ReLU) can be treated as setting a minimal hamming distance threshold between network inputs and weights. This thresholding scheme, on the one hand, can be improved by introducing double-thresholding on both positive and negative extremes of neuron outputs. On the other hand, ReLUs turn out to be *non-essential* and can be removed from networks trained for simple tasks like MNIST classification. The proposed *generalized hamming network* (GHN) as such not only lends itself to rigorous analysis and interpretation within the fuzzy logic theory but also demonstrates fast learning speed, well-controlled behaviour and state-of-the-art performances on a variety of learning tasks.

## 1 Introduction

Since early 1990s the integration of fuzzy logic and computational neural networks has given birth to the *fuzzy neural networks* (FNN) [1]. While the formal fuzzy set theory provides a strict mathematical framework in which vague conceptual phenomena can be precisely and rigorously studied [2, 3, 4, 5], application-oriented fuzzy technologies lag far behind theoretical studies. In particular, fuzzy neural networks have only demonstrated limited successes on some toy examples such as [6, 7]. In order to catch up with the rapid advances in recent neural network developments, especially those with deep layered structures, it is the goal of this paper to demonstrate the relevance of FNN, and moreover, to provide a novel view on its non-fuzzy counterparts.

Our revisiting of FNN is not merely for the fond remembrances of the golden age of "soft computing" [8]. Instead it provides a novel and theoretically justified perspective of neural computing, in which we are able to re-examine and demystify some useful techniques that were proposed to improve either effectiveness or efficiency of neural networks training processes. Among many others, *batch normalization* (BN) [9] is probably the most influential yet mysterious trick, that significantly improved the training efficiency by adapting to the change in the distribution of layers' inputs (coined as *internal covariate shift*). Such kind of adaptations, when viewed within the fuzzy neural network framework, can be interpreted as rectifications to the deficiencies of neuron outputs with respect to the rightful *generalized hamming distance* (see definition 1) between inputs and neuron weights. Once

the appropriate rectification is applied , the ill effects of internal covariate shift are automatically eradicated, and consequently, one is able to enjoy the fast training process without resorting to a sophisticated learning method used by BN.

Another crucial component in neural computing, Rectified linear unit (ReLU), has been widely used due to its strong biological motivations and mathematical justifications [10, 11, 12]. We show that within the *generalized hamming group* endowed with generalized hamming distance, ReLU can be regarded as setting a minimal hamming distance threshold between network input and neuron weights. This novel view immediately leads us to an effective double-thresholding scheme to suppress fuzzy elements in the generalized hamming group.

The proposed *generalized hamming network* (GHN) forms its foundation on the cornerstone notion of *generalized hamming distance* (GHD), which is essentially defined as $h(x, w) := x + w - 2xw$ for any $x, w \in \mathbb{R}$ (see definition 1). Its connection with the inferencing rule in neural computing is obvious: the last term $(-2xw)$ corresponds to element-wise multiplications of neuron inputs and weights, and since we aim to measure the GHD between inputs $x$ and weights $w$, the bias term then should take the value $x + w$. In this article we define any network that has its neuron outputs fulfilling this requirement (3) as a *generalized hamming network*. Since the underlying GHD induces a fuzzy XOR logic, GHN lends itself to rigorous analysis within the fuzzy logics theory (see definition 4). Apart from its theoretical appeals, GHN also demonstrates appealing features in terms of fast learning speed, well-controlled behaviour and simple parameter settings (see Section 4).

## 1.1   Related Work

*Fuzzy logic and fuzzy neural network*: the notion of fuzzy logic is based on the rejection of the fundamental *principle of bivalence* of classical logic i.e. any declarative sentence has only two possible truth values, *true* and *false*. Although the earliest connotation of fuzzy logic was attributed to Aristotle, the founder of classical logic [13], it was Zadeh's publication in 1965 that ignited the enthusiasm about the theory of fuzzy sets [2]. Since then mathematical developments have advanced to a very high standard and are still forthcoming to day [3, 4, 5]. *Fuzzy neural networks* were proposed to take advantages of the flexible knowledge acquiring capability of neural networks [1, 14]. In theory it was proved that fuzzy systems and certain classes of neural networks are equivalent and convertible with each other [15, 16]. In practice, however, successful applications of FNNs are limited to some toy examples only [6, 7].

*Demystifying neural networks*: efforts of interpreting neural networks by means of propositional logic dated back to McCulloch & Pitts' seminial paper [17]. Recent research along this line include [18] and the references therein, in which First Order Logic (FOL) rules are encoded using soft logic on continuous truth values from the interval $[0, 1]$. These interpretations, albeit interesting, seldom explain effective neural network techniques such as batch normalization or ReLU. Recently [19] provided an improvement (and explanation) to batch normalization by removing dependencies in weight normalization between the examples in a minibatch.

*Binary-valued neural network*: Restricted Boltzmann Machine (RBM) was used to model an "ensemble of binary vectors" and rose to prominence in the mid-2000s after fast learning algorithms were demonstrated by Hinton et. al. [20, 21]. Recent binarized neural network [22, 23] approximated standard CNNs by binarizing filter weights and/or inputs, with the aim to reduce computational complexity and memory consumption. The XNOR operation employed in [23] is limited to binary hamming distance and not readily applicable to non-binary neuron weights and inputs.

*Ensemble of binary patterns*: the distributive property of GHD described in (1) provides an intriguing view on neural computing – even though real-valued pattens are involved in the computation, the computed GHD is strictly equivalent to the mean of binary hamming distances across two *ensembles of binary patterns*! This novel view illuminates the connection between generalized hamming networks and efficient binary features, that have long been used in various computer vision tasks, for instance, the celebrated Adaboost face detection[24], numerous binary features for key-point matching [25, 26] and binary codes for large database hashing [27, 28, 29, 30].

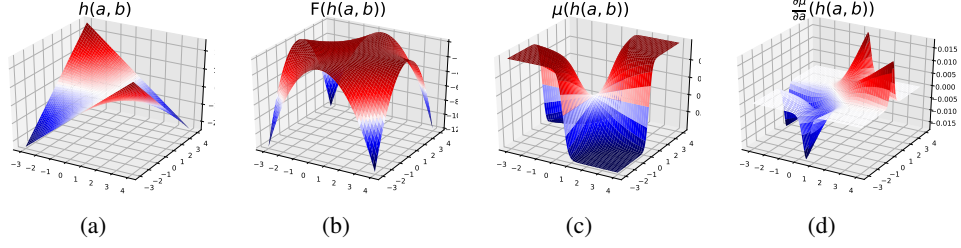

Figure 1: (a) $h(a, b)$ has one *fuzzy region* near the identity element 0.5 (in white), two *positively confident* (in red) and two *negatively confident* (in blue) regions from above and below, respectively. (b) Fuzziness $F(h(a, b)) = h(a, b) \oplus h(a, b)$ has its maxima along $a = 0.5$ or $b = 0.5$. (c) $\mu(h(a, b)) : U \to I$ where $\mu(h) = 1/(1 + exp(0.5 - h))$ is the logistic function to assign membership to fuzzy set elements (see definition 4). (d) partial derivative of $\mu(h(a, b))$. Note that magnitudes of gradient in the fuzzy region is non-negligible.

## 2 Generalized Hamming Distance

**Definition 1.** Let $a, b, c \in U \subseteq \mathbb{R}$, and a *generalized hamming distance* (GHD), denoted by $\oplus$, be a binary operator $h : U \times U \to U$; $\boxed{h(a, b) := a \oplus b = a + b - 2 \cdot a \cdot b}$. Then

(i) for $U = \{0, 1\}$ GHD de-generalizes to binary hamming distance with
$0 \oplus 0 = 0; 0 \oplus 1 = 1; 1 \oplus 0 = 1; 1 \oplus 1 = 0$;

(ii) for $U = [0.0, 1.0]$ the unitary interval $I$, $a \oplus b \in I$ (closure);

*Remark*: this case is referred to as the "restricted" hamming distance, in the sense that inverse of any elements in $I$ are not necessarily contained in $I$ (see below for definition of inverse).

(iii) for $U = \mathbb{R}$, $\mathcal{H} := (\mathbb{R}, \oplus)$ is a group satisfying five *abelian group* axioms, thus is referred to as the *generalized hamming group* or *hamming group*:

- $a \oplus b = (a + b - 2 \cdot a \cdot b) \in \mathbb{R}$ (closure);
- $a \oplus b = (a + b - 2 \cdot a \cdot b) = b \oplus a$ (commutativity);
- $(a \oplus b) \oplus c = (a + b - 2 \cdot a \cdot b) + c - 2(a + b - 2 \cdot a \cdot b)c$
  $= a + (b + c - 2 \cdot b \cdot c) - 2 \cdot a \cdot (b + c - 2 \cdot b \cdot c) = a \oplus (b \oplus c)$ (associativity);
- $\exists e = 0 \in \mathbb{R}$ such that $e \oplus a = a \oplus e = (0 + a - 2 \cdot 0 \cdot a) = a$ (identity element);
- for each $a \in \mathbb{R} \setminus \{0.5\}$, $\exists a^{-1} := a/(2 \cdot a - 1)$ s.t. $a \oplus a^{-1} = (a + \frac{a}{2 \cdot a - 1} - 2a \cdot \frac{a}{2 \cdot a - 1})$
  $= 0 = e$; and we define $\infty := (0.5)^{-1}$ (inverse element).

*Remark*: note that $1 \oplus a = 1 - a$ which *complements* $a$. "0.5" is a fixed point since $\forall a \in \mathbb{R}, 0.5 \oplus a = 0.5$, and $0.5 \oplus \infty = 0$ according to definition[1].

(iv) GHD naturally leads to a measurement of *fuzziness*: $F(a) := a \oplus a, \mathbb{R} \to (-\infty, 0.5] :$ $F(a) \geq 0, \forall a \in [0, 1]; F(a) < 0$ otherwise. Therefore $[0, 1]$ is referred to as the *fuzzy region* in which $F(0.5) = 0.5$ has the maximal fuzziness and $F(0) = F(1) = 0$ are two boundary points. Outer regions $(-\infty, 0]$ and $[1, \infty)$ are negative and positive *confident regions* respectively. See Figure 1 (a) for the surface of $h(a, b)$ which has one central *fuzzy region*, two *positive confident* and two *negative confident* regions.

(v) The *direct sum* of hamming group is still a hamming group $\mathcal{H}^L := \oplus_{l \in L} \mathcal{H}_l$: let $\mathbf{x} = \{x_1, \ldots, x_L\}, \mathbf{y} = \{y_1, \ldots, y_L\} \in \mathcal{H}^L$ be two group members, then the *generalized hamming distance* is defined as the arithmetic mean of element-wise GHD: $\mathcal{G}^L(\mathbf{x} \oplus^L \mathbf{y}) := \frac{1}{L}(x_1 \oplus y_1 + \ldots + x_L \oplus y_L)$.
And let $\tilde{x} = (x_1 + \ldots x_L)/L, \tilde{y} = (y_1 + \ldots y_L)/L$ be arithmetic means of respective elements, then $\boxed{\mathcal{G}^L(\mathbf{x} \oplus^L \mathbf{y}) = \tilde{x} + \tilde{y} - \frac{2}{L}(\mathbf{x} \cdot \mathbf{y})}$, where $\mathbf{x} \cdot \mathbf{y} = \sum_{l=1}^{L} x_l \cdot y_l$ is the dot product.

(vi) *Distributive property*: let $\bar{\mathbf{X}}^M = (\mathbf{x}^1 + \ldots \mathbf{x}^M)/M \in \mathcal{H}^L$ be *element-wise arithmetic mean* of a set of members $\mathbf{x}^m \in \mathcal{H}^L$, and $\bar{\mathbf{Y}}^N$ be defined in the same vein. Then GHD is *distributive*:

$$
\begin{aligned}
\mathcal{G}^L(\bar{\mathbf{X}}^M \oplus^L \bar{\mathbf{Y}}^N) = \frac{1}{L}\sum_{l=1}^{L} \bar{x}_l \oplus \bar{y}_l &= \frac{1}{M}\frac{1}{N}\frac{1}{L}\sum_{m=1}^{M}\sum_{n=1}^{N}\sum_{l=1}^{L} x_l^m \oplus y_l^n \\
&= \frac{1}{MN}\sum_{m=1}^{M}\sum_{n=1}^{N} \mathcal{G}^L(\mathbf{x}^m \oplus^L \mathbf{y}^n).
\end{aligned}
\tag{1}
$$

*Remark*: in case that $x_l^m, y_l^n \in \{0,1\}$ i.e. for two sets of binary patterns, the *mean of binary hamming distance* between two sets can be efficiently computed as the GHD between two real-valued patterns $\bar{\mathbf{X}}^M, \bar{\mathbf{Y}}^N$. Conversely, a real-valued pattern can be viewed as the element-wise average of an ensemble of binary patterns.

## 3 Generalized Hamming Network

Despite the recent progresses in deep learning, artificial neural networks has long been criticized for its "black box" nature: "they capture *hidden* relations between inputs and outputs with a highly accurate approximation, but no definitive answer is offered for the question of how they work" [16]. In this section we provide an interpretation on neural computing by showing that, if the condition specified in (3) is fulfilled, outputs of each neuron can be strictly defined as the generalized hamming distance between inputs and weights. Moreover, the computations of GHD induces fuzzy implication of XOR connective, and therefore, the inferencing of entire network can be regarded as a logical calculus in the same vein as described in McCulloch & Pitts' seminial paper [17].

### 3.1 New perspective on neural computing

The bearing of *generalized hamming distance* on neural computing is elucidated by looking at the negative of generalized hamming distance, (GHD, see definition 1), between inputs $\mathbf{x} \in \mathcal{H}^L$ and weights $\mathbf{w} \in \mathcal{H}^L$ in which $L$ denotes the length of neuron weights e.g. in convolution kernels:

$$
-\mathcal{G}^L(\mathbf{w} \oplus^L \mathbf{x}) = \frac{2}{L}\mathbf{w} \cdot \mathbf{x} - \frac{1}{L}\sum_{l=1}^{L} w_l - \frac{1}{L}\sum_{l=1}^{L} x_l
\tag{2}
$$

Divide (2) by the constant $\frac{2}{L}$ and let

$$
\boxed{b = -\frac{1}{2}\Big(\sum_{l=1}^{L} w_l + \sum_{l=1}^{L} x_l\Big)}
\tag{3}
$$

then it becomes the familiar form $(\mathbf{w} \cdot \mathbf{x} + b)$ of neuron outputs save the non-linear activation function. By enforcing the bias term to take the given value in (3), standard neuron outputs measure negatives of GHD between inputs and weights. Note that, for each layer, the bias term $\sum_{l=1}^{L} x_l$ is averaged over neighbouring neurons in individual input image. The bias term $\sum_{l=1}^{L} w_l$ is computed separately for each filter in fully connected or convolution layers. When weights are updated during the optimization, $\sum_{l=1}^{L} w_l$ changes accordingly to keep up with weights and maintain stable neuron outputs. We discuss below (re-)interpretations of neural computing in terms of GHD.

**Fuzzy inference:** As illustrated in definition 4 GHD induces a fuzzy XOR connective. Therefore the negative of GHD quantifies the *degree of equivalence* between inputs $\mathbf{x}$ and weights $\mathbf{w}$ (see definition 4 of fuzzy XOR), i.e. the fuzzy truth value of the statement "$\mathbf{x} \leftrightarrow \mathbf{w}$" where $\leftrightarrow$ denotes a fuzzy equivalence relation. For GHD with multiple layers stacked together, neighbouring neuron outputs from the previous layer are integrated to form composite statements e.g. "$(\mathbf{x}_1^1 \leftrightarrow \mathbf{w}_1^1, \ldots, \mathbf{x}_i^1 \leftrightarrow \mathbf{w}_i^1) \leftrightarrow \mathbf{w}_j^2$" where superscripts correspond to two layers. Thus stacked layers will form more complex, and hopefully more powerful, statements as the layer depth increases.

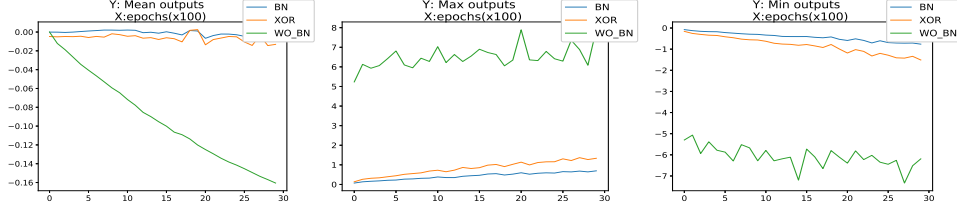

Figure 2: Left to right: mean, max and min of neuron outputs, with/without batch normalized (BN, WO_BN) and generalized hamming distance (XOR). Outputs are averaged over all 64 filters in the first convolution layer and plotted for 30 epochs training of a MNIST network used in our experiment (see Section 4).

**Batch normalization demystified:** When a mini-batch of training samples $\mathbf{X} = \{\mathbf{x}^1, \ldots, \mathbf{x}^M\}$ is involved in the computation, due to the distributive property of GHD, the data-dependent bias term $\sum_{l=1}^{L} x_l$ equals the arithmetic mean of corresponding bias terms computed for each sample in the mini-batch i.e. $\frac{1}{M} \sum_{m=1}^{M} \sum_{l=1}^{L} x_l^m$. It is almost impossible to maintain a constant scalar $b$ that fulfils this requirement when mini-batch changes, especially at deep layers of the network whose inputs are influenced by weights of incoming layers. The celebrated batch normalization (BN) technique therefore proposed a learning method to compensate for the input vector change, with additional parameters $\gamma, \beta$ to be learnt during the training [9]. It is our conjecture that batch normalization is approximating these rightful bias through optimization, and this connection is empirically revealed in Figure 2 with very similar neuron outputs obtained by BN and GHD. Indeed they are highly correlated during the course of training (with Pearson correlation coefficient=0.97), confirming our view that BN is attempting to influence the bias term according to (3).

Once $b$ is enforced to follow (3), *neither the optimization of bias terms nor the sophisticated learning method of BN is needed*. In the following section we will illustrate a rectified neural network designed as such.

**Rectified linear units (ReLU) redesigned:** Due to its strong biological motivations [10] and mathematical justifications [11], rectified linear unit (ReLu) is the most popular activation function used for deep neural network [31]. If neuron outputs are rectified as the generalized hamming distances, the activation function $max(0, 0.5 - h(\mathbf{x}, \mathbf{w}))$ then simply sets a minimal hamming distance threshold of $0.5$ (see Figure 1). Astute readers may immediately spot two limitations of this activation function: a) it only takes into account the negative confidence region while disregards positive confidence regions; b) it allows elements in the fuzzy regime near $0.5$ to misguide the optimization with their non-negligible gradients.

A straightforward remedy to ReLU is to suppress elements within the fuzzy region by setting outputs between $[0.5 - r, 0.5 + r]$ to $0.5$, where $r$ is a parameter to control acceptable fuzziness in neuron outputs. In particular, we may set thresholds adaptively e.g. $[0.5 - r \cdot O, 0.5 + r \cdot O]$ where $O$ is the maximal magnitude of neuron outputs and the threshold ratio $r$ is adjusted by the optimizer. This *double-thresholding* strategy effectively prevents noisy gradients of fuzzy elements, since $0.5$ is a fixed point and $x \oplus 0.5 = 0.5$ for any $x$. Empirically we found this scheme, in tandem with the rectification (3), dramatically boosts the training efficiency for challenging tasks such as CIFAR10/100 image classification. It must be noted that, however, the use of non-linear activation as such is *not essential* for GHD-based neural computing. When the double-thresholding is switched-off (by fixing $r = 0$), the learning is prolonged for challenging CIFAR10/100 image classification but its influence on the simple MNIST classification is almost negligible (see Section 4 for experimental results).

## 3.2 Ganeralized hamming network with induced fuzzy XOR

**Definition 2.** A *generalized hamming network* (GHN) is any networks consisting of neurons, whose outputs $\mathbf{h} \in \mathcal{H}^L$ are related to neuron inputs $\mathbf{x} \in \mathcal{H}^L$ and weights $\mathbf{w} \in \mathcal{H}^L$ by $\boxed{\mathbf{h} = \mathbf{x} \oplus^L \mathbf{w}}$.

*Remark*: In case that the bias term is computed directly from (3) such that $\mathbf{h} = \mathbf{x} \oplus^L \mathbf{w}$ is fulfilled strictly, the network is called a *rectified GHN* or simply a *GHN*. In other cases where bias terms are approximating the rightful offsets (e.g. by batch normalization [9]), the trained network is called an *approximated GHN*.

Compared with traditional neural networks, the optimization of bias terms is no longer needed in GHN. Empirically, it is shown that the proposed GHN benefits from a fast and robust learning process that is on par with that of the batch-normalization approach, yet without resorting to sophisticated learning process of additional parameters (see Section 4 for experimental results). On the other hand, GHN also benefits from the rapid developments of neural computing techniques, in particular, those employing parallel computing on GPUs. Due to this efficient implementation of GHNs, it is the first time that fuzzy neural networks have demonstrated state-of-the-art performances on learning tasks with large scale datasets.

Often neuron outputs are clamped by a logistic activation function to within the range $[0, 1]$, so that outputs can be compared with the target labels in supervised learning. As shown below, GHD followed by such a non-linear activation actually induces a fuzzy XOR connective. We briefly review basic notion of fuzzy set used in our work and refer readers to [2, 32, 13] for thorough treatments and review of the topic.

**Definition 3. Fuzzy Set:** Let $X$ be an universal set of elements $x \in X$, then a *fuzzy set $A$* is a set of pairs: $A := \{(x, \mu_A(x)) | x \in X, \mu_A(x) \in I\}$, in which $\mu_A : X \to I$ is called the membership function (or grade membership).

*Remark*: In this work we let $X$ be a Cartesian product of two sets $X = P \times U$ where $P$ are (2D or 3D) collection of neural nodes and $U$ are real numbers in $\subseteq I$ or $\subseteq R$. We define the membership function $\mu_X(x) := \mu_U(x_p), \forall x = (p, x_p) \in X$ such that it is dependent on $x_p$ only. For the sake of brevity we abuse the notation and use $\mu(x)$, $\mu_X(x)$ and $\mu_U(x_p)$ interchangeably.

**Definition 4. Induced fuzzy XOR**: let two fuzzy set elements $a, b \in U$ be assigned with respective grade or membership by a membership function $\mu : U \to I : \mu(a) = i, \mu(b) = j$, then the generalized hamming distance $h(a, b) : U \times U \to U$ induces a fuzzy XOR connective $E : I \times I \to I$ whose membership function is given by

$$\mu_R(i, j) = \mu(h(\mu^{-1}(i), \mu^{-1}(j))). \tag{4}$$

*Remark*: For the restricted case $U = I$ the membership function can be trivially defined as the identity function $\mu = \mathrm{id}_I$ as proved in [4].

*Remark*: For the generalized case where $U = \mathbb{R}$, the fuzzy membership $\mu$ can be defined by a sigmoid function such as logistic, *tanh* or any function $: U \to I$. In this work we adopt the logistic function $\mu(a) = \frac{1}{1 + \exp(0.5 - a)}$ and the resulting fuzzy XOR connective is given by following membership function:

$$\mu_R(i, j) = \frac{1}{1 + \exp\left(0.5 - \mu^{-1}(i) \oplus \mu^{-1}(j)\right)}, \tag{5}$$

where $\mu^{-1}(a) = -\ln(\frac{1}{a} - 1) + \frac{1}{2}$ is the inverse of $\mu(a)$. Following this analysis, it is possible to rigorously formulate neuron computing of the entire network according to inference rules of fuzzy logic theory (in the same vein as illustrated in [17]). Nevertheless, research along this line is out of the scope of the present article and will be reported elsewhere.

## 4 Performance evaluation

### 4.1 A case study with MNIST image classification

**Overall performance**: we tested a simple four-layered GHN (cv[1,5,5,16]-pool-cv[16,5,5,64]-pool-fc[1024]-fc[1024,10]) on the MNIST dataset with $99.0\%$ test accuracy obtained. For this relatively simple dataset, GHN is able to reach test accuracies above 0.95 with 1000 mini-batches and a learning rate 0.1. This learning speed is on par with that of the batch normalization (BN), but without resorting to the learning of additional parameters in BN. It was also observed a wide range of large learning rates (from 0.01 to 0.1) all resulted in similar final accuracies (see below). We ascribe this well-controlled robust learning behaviour to rectified bias terms enforced in GHNs.

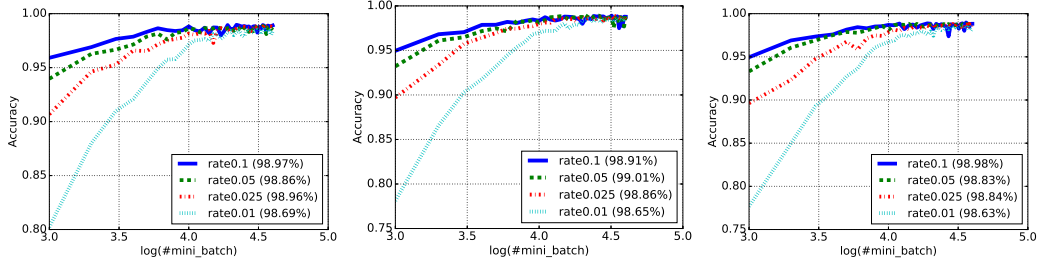

Figure 3: Test accuracies of MNIST classification with Generalized Hamming Network (GHN). Left: test accuracies without using non-linear activation (by setting $r = 0$). Middle: with $r$ optimized for each layer. Right: with $r$ optimized for each filter. Four learning rates i.e. $\{0.1, 0.05, 0.025, 0.01\}$ are used for each case with the final accuracy reported in brackets. Note that the number of mini-batch are in logarithmic scale along x-axis.

*Influence of learning rate*: This experiment compares performances with different learning rates and Figure 3 (middle,right) show that a very large learning rate (0.1) leads to much faster learning without the risk of divergences. A small learning rate (0.01) suffice to guarantee the comparable final test accuracy. Therefore we set the learning rate to a constant 0.1 for all experiments unless stated otherwise.

*Influence of non-linear double-thresholding*: The non-linear double-thresholding can be turned off by setting the threshold ratio $r = 0$ (see texts in Section 3.1). Optionally the parameter $r$ is automatically optimized together with the optimization of neuron weights. Figure 3 (left) shows that the GHN without non-linear activation (by setting $r = 0$) performs equally well as compared with the case where $r$ is optimized (in Figure 3 left, right). There are no significant differences between two settings for this relative simple task.

## 4.2 CIFAR10/100 image classification

In this experiment, we tested a six-layered GHN (cv[3,3,3,64]-cv[64,5,5,256]-pool-cv[256,5,5,256]-pool-fc[1024]-fc[1024,512]-fc[1024,nclass]) on both CIFAR10 (nclass=10) and CIFAR100 (nclass=100) datasets. Figure 4 shows that the double-thresholding scheme improves the learning efficiency dramatically for these challenging image classification tasks: when the parameter $r$ is optimized for each feature filter the numbers of iterations required to reach the same level of test accuracy are reduced by 1 to 2 orders of magnitudes. It must be noted that performances of such a simple generalized hamming network (89.3% for CIFAR10 and 60.1% for CIFAR100) are on par with many sophisticated networks reported in [33]. In our view, the rectified bias enforced by (3) can be readily applied to these sophisticated networks, although resulting improvements may vary and remain to be tested.

## 4.3 Generative modelling with Variational Autoencoder

In this experiment, we tested the effect of rectification in GHN applied to a generative modelling setting. One crucial difference is that the objective is now to minimize reconstruction error instead of classification error. It turns out the double-thresholding scheme is no longer relevant for this setting and thus not used in the experiment.

The baseline network (784-400-400-20) used in this experiment is an improved implementation [34] of the influential paper [35], trained on the MNIST dataset of images of handwritten digits. We have rectified the outputs following (3) and, instead of optimizing the lower bound of the log marginal likelihood as in [35], we directly minimize the reconstruction error. Also we did not include weights regularization terms for the optimization as it is unnecessary for GHN. Figure 5 (left) illustrates the reconstruction error with respect to number of training steps (mini-batches). It is shown that the rectified generalized hamming network converges to a lower minimal reconstruction error as compared to the baseline network, with about 28% reduction. The rectification also leads to a faster convergence, which is in accordance with our observations in other experiments.

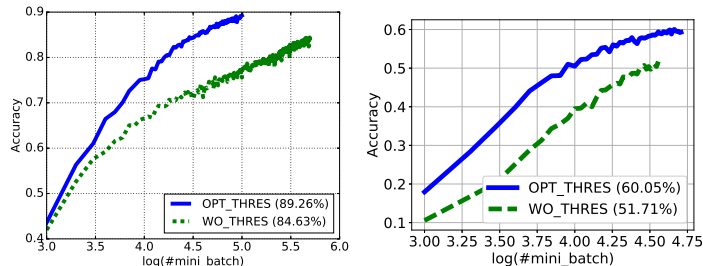

Figure 4: Left: GHN test accuracies of CIFAR10 classification (OPT THRES: parameter $r$ optimized; WO THRES: without nonlinear activation). Right: GHN test accuracies of CIFAR100 classification(OPT THRES: parameter $r$ optimized; WO THRES: without non-linear activation).

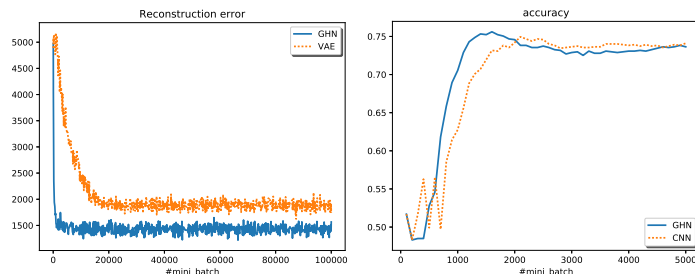

Figure 5: Left: Reconstruction errors of convolution VAE with and w/o rectification. Right: Evaluation accuracies of Sentence classification with GHN rectification and w/o rectification).

## 4.4 Sentence classification

A simple CNN has been used for sentence-level classification tasks and excellent results were demonstrated on multiple benchmarks [36]. The baseline network used in this experiment is a re-implementation of [36] made available from [37]. Figure 5 (right) plots accuracy curves from both networks. It was observed that the rectified GHN did improve the *learning speed*, but did not improve the final accuracy as compared with the baseline network: both networks yielded the final evaluation accuracy around $74\%$ despite that the training accuracy were almost $100\%$. The over-fitting in this experiment is probably due to the relatively small Movie Review dataset size with 10,662 example review sentences, half positive and half negative.

## 5 Conclusion

In summary, we proposed a rectified *generalized hamming network* (GHN) architecture which materializes a re-emerging principle of fuzzy logic inferencing. This principle has been extensively studied from a theoretic fuzzy logic point of view, but has been largely overlooked in the practical research of ANN. The rectified neural network derives fuzzy logic implications with underlying *generalized hamming distances* computed in neuron outputs. Bearing this rectified view in mind, we proposed to compute bias terms analytically without resorting to sophisticated learning methods such as batch normalization. Moreover, we have shown that, the rectified linear units (ReLU) was theoretically non-essential and could be skipped for some easy tasks. While for challenging classification problems, the double-thresholding scheme did improve the learning efficiency significantly.

The simple architecture of GHN, on the one hand, lends itself to being analysed rigorously and this follow up research will be reported elsewhere. On the other hand, GHN is the first fuzzy neural network of its kind that has demonstrated fast learning speed, well-controlled behaviour and state-of-the-art performances on a variety of learning tasks. By cross-checking existing networks against GHN, one is able to grasp the most essential ingredient of deep learning. It is our hope that this kind of comparative study will shed light on future deep learning research and eventually open the "black box" of artificial neural networks [16].

## Acknowledgement

I am grateful to anonymous reviewers for their constructive comments to improve the quality of this paper. I greatly appreciate valuable discussions and supports from colleagues at Nokia Technologies.

## Footnotes

[1] By this extension, it is $\overline{\overline{\mathbb{R}}} = \mathbb{R} \cup \{-\infty, +\infty\}$ instead of $\mathbb{R}$ on which we have all group members.

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
