[Reviews · NeurIPS 2017]

Reviewer 1



The authors use a notion of generalized hamming distance, to shed light on the success of Batch normalization and ReLU units. After reading the paper, I am still very confused about its contribution. The authors claim that generalized hamming distance offers a better view of batch normalization and relus, and explain that in two paragraphs in pages 4,5. The explanation for batch normalization is essentially contained in the following phrase: “It turns out BN is indeed attempting to compensate for deficiencies in neuron outputs with respect to GHD. This surprising observation indeed adheres to our conjecture that an optimized neuron should faithfully measure the GHD between inputs and weights.” I do not understand how this is explaining the effects or performance of batch normalization. The authors then propose a generalized hamming network, and suggest that "it demystified and confirmed effectiveness of practical techniques such as batch normalization and ReLU". Overall, this is a poorly written paper, with no major technical contribution, or novelty, and does not seem to provide any theoretical insights on the effectiveness of BN or ReLUs. Going beyond the unclear novelty and technical contribution, the paper is riddled with typos, grammar and syntax mistakes (below is a list from just the abstract and intro). This is a clear rejection. Typos and grammar/syntax mistakes: —— abstract —— generalized hamming network (GNN) -> generalized hamming network (GHN) GHN not only lends itself to rigiour analysis -> GHN not only lends itself to rigorous analysis “but also demonstrates superior performances” -> but also demonstrates superior performance —— —— intro —— “computational neutral networks” -> computational neural networks “has given birth” -> have given birth “to rectifying misunderstanding of neural computing” -> not sure what the authors are trying to say Once the appropriate rectification is applied , -> Once the appropriate rectification is applied, the ill effects of internal covariate shift is automatically eradicated -> the ill effects of internal covariate shift are automatically eradicated The resulted learning process -> The resulting learning process lends itself to rigiour analysis -> lends itself to rigorous analysis the flexaible knowledge -> the flexible knowledge are equivalent and convertible with other -> are equivalent and convertible with others, or other architectures? successful applications of FNN -> successful applications of FNNs

Reviewer 2



This is a beautiful paper that interprets batch normalization and relu in terms of generalized hamming distance network and develops variations that improves. This connection is surprising (esp fig 2) and very intereting. The correspondence between ghd and bn is interesting. However, it is also a bit non-obvious why this is true. It seems to me that the paper claims that in practice the estimated bias is equal to sum_w and sum_x in (3) and the whole wx+b equates the ghd. However, is it on avearge across all nodes in one layer? Also how does it vary across layers? Does the ghd mainly serve to ensure that there is no information loss going from x to w.? It is a little hard to imagine why we want layers of ghd stacked together. Any explanation in this direction could be helpful. A minor thing: typo in the crucial box in line 80 Overall a great paper. ---------------------- After author rebuttal: Many thanks for the rebuttal from the authors. My scores remain the same. Thanks for the beautiful paper!

Reviewer 3



This paper explores generalized hamming distance in the context of fuzzy neural networks. The paper shows that the neural computation at each neuron can be viewed as calculating the generalized hamming distance between the corresponding weight vector and input vector. This view point is useful as it automatically gives candidate bias parameter at the neuron and learning of additional parameters (using batch normalization) is not required during the training. Furthermore, this view also allows the authors to understand the role of rectified linear units and propose a double-thresholding scheme to make the training process fast and stable. Based on these two observations, the paper proposes generalized hamming networks (GHN) where each neuron exactly computes the generalized hamming distance. These networks also utilize non-linear activations. The paper then presents simulation results showcasing the performance of the proposed GHN for image classification and auto-encoder design. The simulation results demonstrate that GHN have fast learning process and achieve very good performances on the underlying tasks. Interestingly, the simulation results show that the GHN do not necessarily require max-pooling. The paper is well written and explains the main ideas in a clear manner. The simulation results are also convincing. However, it appears that the plots are not complete, e.g., Fig. 4.2 does not show a curve for learning rate = 0.0001. Even though I would have preferred a more comprehensive (both theoretical and practical) evaluations, I over all find the idea of GHN interesting. Minor comments: 1) In abstract, '(GNN)' should be '(GHN)'. 2) In line 55 of page 2, 'felxaible' should be 'flexible'.